# Impaired Cardiac Sympathetic Innervation Increases the Risk of Cardiac Events in Heart Failure Patients with Left Ventricular Hypertrophy and Mechanical Dyssynchrony

**DOI:** 10.3390/jcm10215047

**Published:** 2021-10-28

**Authors:** Takahiro Doi, Tomoaki Nakata, Takahiro Noto, Tomohiro Mita, Daigo Nagahara, Satoshi Yuda, Akiyoshi Hashimoto

**Affiliations:** 1Department of Cardiology, Teine Keijinkai Hospital, Sapporo 006-8555, Japan; nototakahiro1125@yahoo.co.jp (T.N.); mitatomohiro@gmail.com (T.M.); daihagahar@yahoo.co.jp (D.N.); yuda410902@gmail.com (S.Y.); 2Department of Cardiology, Hakodate Goryokaku Hospital, Hakodate 040-8611, Japan; tnakata@sapmed.ac.jp; 3Department of Cardiovascular, Renal and Metabolic Medicine, Sapporo Medical University, Sapporo 060-8543, Japan; akiyys@yahoo.co.jp

**Keywords:** heart failure, cardiac sympathetic function, prognosis, mechanical left ventricular dyssynchrony

## Abstract

Background. Left ventricular mechanical dyssynchrony (LVMD), left ventricular hypertrophy, and impaired cardiac sympathetic innervation are closely related to the development of heart failure (HF) and unfavorable outcomes. Methods and Results. A total of 705 consecutive HF patients with reduced left ventricular ejection fraction (EF) < 50% were registered in our hospital HF database. LVMD and left ventricular mass index (LVMI) were evaluated three-dimensionally by gated myocardial perfusion SPECT. LVMD was measured as a heterogeneity index (phase SD) of the regional contraction phase angles calculated by Fourier analysis. Cardiac sympathetic innervation was quantified as a normalized heart-to-mediastinum ratio (HMR) of the ^123^I-metaiodobenzylguanidine (MIBG) activity. The patients were followed up with a primary end point of lethal cardiac events (CEs) for 42 months. CEs were documented in 246 of the HF patients who had a greater phase SD, greater LVMI, and lower MIBG HMR than those in HF patients without CEs. In the overall multivariate analysis, phase SD, LVMI, and MIBG HMR were identified as significant CE determinants. The three biomarkers were incrementally related to increases in CE risks. Conclusions. Assessment of cardiac sympathetic innervation can further stratify patients with systolic heart failure at increased cardiac risk identified by left ventricular hypertrophy and mechanical dyssynchrony.

## 1. Introduction

Management of patients with heart failure (HF) is becoming a critical concern to be resolved due to rapid increases in the number of elderly patients, and the increase in expenditure for life-saving measures and subsequent long-term medical care [1,2]. Despite the recent advances in pharmacological and non-pharmacological treatments, cardiac mortality remains high, with a 5-year survival rate of only 25% after hospitalization for HF with reduced left ventricular ejection fraction (LVEF) [1,2,3,4,5]. Among the known prognostic clinical variables, left ventricular hypertrophy (LVH) is a classical but still important variable, but has been conventionally assessed by electrocardiography or two-dimensional echocardiography. Attention has recently been given to left ventricular mechanical dyssynchrony (LVMD) in relation not only to prognostic implications, but also to the clinical indication of cardiac re-synchronization treatment (CRT). Despite improvements in cardiac prognosis using cardiac implantable electronic devices, there are still limitations in actual prognostic efficacies [6,7,8]. Besides technical issues of device implantation, this is probably because of the limited assessment using an electrical and/or two-dimensional echo data, resulting in an ineffective CRT rate of nearly 30% [9,10]. On the other hand, three-dimensional evaluation of LVMD and LVH can be done routinely in cardiac nuclear imaging, and our previous studies have shown the improvement of prognostic assessment using three-dimensional nuclear techniques, contributing to optimization of the selection of CRT candidates [11,12,13].

The autonomic nervous system plays crucial roles in the maintenance and compensation of cardiac function and, in contrast, in the progression and fatal outcomes of HF [14,15]. Neuroimaging with a norepinephrine analogue, iodinated metaiodobenzylguanidine (MIBG), enables non-invasive delineation and evaluation of pre-synaptic sympathetic innervation in human hearts using a conventional imaging technique [16,17]. A number of investigations have demonstrated independent and incremental prognostic implications of impaired cardiac sympathetic nerve function assessed by cardiac MIBG imaging in HF patients. In particular, multicenter studies established the prognostic values of cardiac MIBG activity in the risk-stratification and the prediction of cardiac outcomes in HF patients, even when conventional prognostic markers such as NYHA functional class, LVEF, and plasma brain natriuretic peptide (BNP) are considered [18,19,20].

In these contexts, a comprehensive understanding and assessment of the pathophysiological conditions such as three-dimensional LVMD and LVH and alterations of cardiac sympathetic innervation are expected to make the prediction and prevention of lethal cardiac outcomes more accurate in HF patients. In this study, we tested the hypothesis that the integration of three-dimensional LVMD and LVH into the impairment of cardiac sympathetic innervation enables more precise evaluation of cardiac risks in HF patients who are conventionally risk-stratified.

## 2. Methods

### 2.1. Study Patients

A total of 705 patients were consecutively and prospectively registered in our hospital HF database between April 2011 and December 2016 using the following entry criteria: symptomatic HF established by typical symptoms and signs based on the Framingham criteria including palpitation, dyspnea/orthopnea, neck vein distension, peripheral and/or lung congestion S3/S4 gallop, chest radiographic findings of congestive heart failure, echocardiographic LVEF less than 50% at admission, and a significant increase in BNP. Electrocardiographic and imaging examinations were conducted immediately after admission to establish the diagnosis by excluding non-cardiac diseases with similar symptoms or signs. HF etiologies, such as ischemic versus non-ischemic, were also differentiated using electrocardiographic and cardiac imaging studies with or without a stress approach and coronary angiographic examination depending on the patient’s clinical conditions. The following patients were excluded from the analysis: patients aged less than 20 years; patients who refused entry to the study or refused resuscitation treatment; and patients with overt serious non-cardiac disorders such as malignancy, infections, or hemorrhagic diseases. Just before discharge, blood examinations of hemoglobin (Hb), BNP, and creatinine were performed and the estimated glomerular filtration rate (eGFR) was calculated. The plasma BNP level was measured in the initial 310 patients (44%) then and NT-pro BNP level was measured in the next 395 patients (56%). Because of the two different methods, BNP/NT-pro BNP staging was performed on the basis of ESC heart failure guidelines as follows: 0–40 pg/mL and 0–125 pg/mL for stage 1, 41–100 pg/mL and 126–400 pg/mL for stage 2, 101–200 pg/mL and 401–900 pg/mL for stage 3, and 201– pg/mL and 901– pg/mL for stage 4.

All participants provided written informed consent. The study protocol conformed to the ethical guidelines of the 1975 Declaration of Helsinki, as reflected in a priori approval by our institution’s human research committee.

### 2.2. Two-Dimensional Echocardiographic Assessment

Two-dimensional echocardiography was performed from standard parasternal long-axis and apical four-, three-, and two-chamber views in a left lateral decubitus position using commercially available ultrasound machines equipped with a 2.5-MHz variable frequency transducer by echocardiographic technicians, repeatedly when necessary. The following echocardiographic parameters measured before discharge were used for the analysis: left atrium diameter (LAD, mm); left ventricular end-diastolic diameter (LVDd, mm); septal wall thickness at end-diastole (IVSTd, mm); posterior wall thickness at end-diastole (PTWd, mm); LVEF (%), calculated using the biplane modified Simpson’s method; left ventricular volume at end-diastole (EDV, ml); left ventricular volume at end-systole (ESV, mL); E wave velocity (m/sec); and left ventricular deceleration time of E wave (Dct, msec) and septal E/e′.

### 2.3. Assessment of LVMD and Left Ventricular Mass Index

LVMD and left ventricular mass index (LVMI) as an index of LVH were evaluated using three-dimensional data derived from resting myocardial perfusion imaging (MPI) with ^99m^Tc-tetrofosmin of 300 MBq, as shown by our previous study (Figure 1) [11,12,13]. Briefly, MPI SPECT was performed using an ECG-gated approach with a frame rate of 16 and a single-head gamma camera equipped with a high-resolution, parallel-hole collimator. LVMD and LVMI were measured using three-dimensionally reconstructed ECG-gated data of serial 10- to 14-short-axis SPECT images with the Heart Function View software (HFV version 1.1;Nihon MediPhysics, Tokyo, Japan) [12,13] and Corridor 4DM version 6.0 software (INVIA Medical Imaging Solutions, Ann Arbor, MI, USA) [11], respectively. LVMD was quantified as a standard deviation (SD) of phase angles (degrees) on a phase histogram (Figure 1). Phase angles at the initiation of systole were calculated mathematically by a high-ordered Fourier analysis of myocardial count changes (i.e., time−activity curve) in each pixel during a cardiac cycle, which is basically identical to a regional time−volume curve. The left ventricular mass was normalized as LVMI (g/m^2^) using body surface area.

### 2.4. Cardiac ^123^I-MIBG Imaging

In a stabilized patient’s condition, cardiac imaging with ^123^I-MIBG of 111 MBq was performed using a gamma camera equipped with a low-energy, general-purpose collimator in a fasting and resting condition for 15–30 min (early image) and 4 h (late image) after an intravenous tracer injection, as described previously [11,12,13,14]. Cardiac ^123^I-MIBG activity was measured as a heart/mediastinum ratio (HMR) by a region of interest being automatically set on the upper mediastinum and the whole cardiac region on a planar anterior image with dedicated MIBG software (smart MIBG software, Tokyo, Japan) operated by an experienced nuclear medicine technician without knowledge of any clinical data. ^123^I-MIBG washout kinetics from the heart was calculated as the washout rate (WR) from early and late cardiac ^123^I-MIBG activities without a decay correction. The cardiac MIBG HMR was standardized to the medium-energy collimator condition by the mathematical method established by cross-calibration phantom experiments [18]. This method can minimize variations in MIBG HMR originating from different collimators such as low- and medium-energy ones between institutions or study populations.

### 2.5. Follow-Up Protocol

Patients were regularly followed up at an out-patient care unit by cardiologists for one year or more when the patients survived. The primary endpoint was lethal cardiac events including sudden cardiac death, death due to progression of pump failure, life-threatening lethal ventricular tachyarrhythmias, and appropriate ICD therapy. Clinical outcomes were confirmed retrospectively by reviewing medical records. Sudden cardiac death was defined as witnessed cardiac arrest and death within 1 h after acute onset of symptoms or unexpected death in patients being well within the previous 24 h. This study was based on the principles outlined in the Declaration of Helsinki, and informed consent for enrollment in our database and usage for clinical study was obtained according to the guidelines of the ethics committee of our hospital.

### 2.6. Statistical Analysis

A statistical value is shown as mean ± 1 SD. Mean values were compared between the two groups using the unpaired t-test, and categorical variables were compared using the Chi-square test. The Kaplan−Meier analysis using the key parameters identified in this study was used to create a time-dependent, cumulative event-free curve, and the log-rank test was also used for comparison of the curves, if necessary. Following univariate analysis, multivariate analysis with a Cox hazard proportional model was performed using the statistically appropriate number of significant variables identified to calculate the hazard ratios and 95% confidence intervals (CIs). A receiver operating characteristic (ROC) analysis was performed to determine an optimal cutoff value of an independent significant parameter for differentiating a risk category. Global Chi-square values were calculated when incremental prognostic values were needed to be clarified, using several significant variables identified above. A computer software program, SAS for Windows, version 9.4 (SAS Institute, Cary, NC, USA), was used for these analyses. A *p*-value less than 0.05 was considered significant.

## 3. Results

The patients included 517 males (73.3%) and had a mean age of 67.1 ± 12.6 years and a mean LVEF of 35.7 ± 10.6%. During a mean follow-up period of 42.2 months, lethal cardiac events were documented in 246 (35%) of the patients: 190 died due to HF progression, 22 had lethal ventricular arrhythmias, 22 had sudden cardiac death, and 12 had appropriate ICD shock treatment against lethal ventricular arrhythmias. Patients with cardiac events were older and leaner, had a greater NYHA functional class, more frequently had a history of lethal ventricular tachyarrhythmias, and had lower levels of eGFR and hemoglobin when compared with those without cardiac events (Table 1).

Figure 1 shows the measurements of the phase SD and MIBG HMR in two typical cases. Case 1 with LVEF of 32% had nearly normal LVMI (97 g/m^2^) and HMR (2.66), and a relatively small phase SD (19.0 degrees), and had no cardiac event during the follow-up period (Figure 1, Case 1). In contrast, Case 2 had markedly reduced LVEF (28%) and HMR (1.48), had increased LVMI (178 g/m^2^) and phase SD (40 degrees), and the patient died of progression of HF (Figure 1, Case 2). In addition to echocardiographic functional abnormalities, patients with cardiac events had both greater phase SD and LVMI and lower HMR than did those without cardiac events: phase SD, 37.4 ± 10.8 vs. 33.1 ± 9.9 degrees, *p* < 0.0001; LVMI, 129.3 ± 35.7 vs. 117.9 ± 28.7 g/m^2^, *p* < 0.0001; MIBG HMR, 1.71 ± 0.39 vs. 1.99 ± 0.48, *p* < 0.0001 (Table 2). In the multivariate Cox analysis, phase SD, LVMI, and MIBG HMR were identified as significant independent variables, as well as age, NYHA functional class, BNP/NT-pro BNP, eGFR, amiodarone use, and LV ESV (Table 3).

Using cutoff values determined by the ROC analysis (Appendix A), high-risk HF categories were clearly discriminated from low-risk categories as follows. Patients with a phase SD greater than 37 degrees, MIBG HMR less than 1.89, or LVMI more than 122.9 g/m^2^ had significantly lower event-free rates than the other patients (Figure 2). The combined use of LVMI or phase SD with MIBG HMR more clearly discriminated patients with greater CV risks from others (Figure 3). LVMI > 122.9 g/m^2^, Phase SD > 37 degrees, and MIBG HMR < 1.89 were incrementally related to increases in CV rates, leading to the lowest CV-free curve when all of the three abnormalities were accumulated (Figure 4).

Furthermore, when analyzed separately in heart failure groups with male and female, age (65 years old older, or younger), LVEF (reduced EF (EF < 40%) or mid-range EF (EF ≥ 40%)), the similar Kaplan-Meier event-free curves were obtained, respectively. (Appendix A).

When analyzed separately in heart failure groups with and without ischemic etiology, similar Kaplan−Meier event-free curves were obtained, strongly indicating the critical values of the accumulation of the three prognostic variables (Figure 5).

When the significant multivariate variables, including clinical prognostic biomarkers (Table 3), were combined, LVMI, phase SD, and MIBG HMR showed synergistically increased prognostic values maximally, with a Chi-square value of 188.8 (Figure 4).

## 4. Discussion

Impairment of cardiac sympathetic innervation was shown to be independently and incrementally related to lethal cardiac outcomes in combination with three-dimensional LVMI and LVMD, contributing to a further improvement in the risk-stratification of patients with HFrEF evaluated conventionally.

### 4.1. Assessment of LVMI and LVMD

The presented findings strengthened the prognostic values of LVMI and LVMD, which are already known. Methodologically, there are several rationales for the use of three-dimensional data for LVMI and LVMD. Morphological and functional left ventricular remodeling occurs globally and is responsible for the manifestation and exacerbation of HF, leading to lethal outcomes. Electrical remodeling assessed by QRS complex duration and two-dimensional assessment are surrogated but limited, as in this point of view. In addition to conventional cardiac functional parameters, the recent widely available technology that was used in this study enabled semi-automatic quantification of LVMI and LVMD at a routine nuclear cardiology practice, without any additional cost and imaging time. These advances in three-dimensional analysis enable more precise risk-stratification of HF patients and overcome the technical limitations of conventional methods [9,10,11], possibly contributing to a more cost-effective diagnostic strategy in combination with an assessment of myocardial ischemia and viability for the differentiation of HF etiology [4].

### 4.2. Assessment of Cardiac Sympathetic Innervation

The prognostic values of cardiac neuronal remodeling assessed by the cardiac ^123^I-MIBG activity have been consistently shown in the past two decades [16,17,20]. It has, however, been desirable to standardize the quantitative method for cardiac ^123^I-MIBG activity as HMR to cancel the effects due to methodological variations such as collimators used in facilities [18,19,20]. It is important to compare MIBG data among different facilities or among different investigations. The method presented for the standardization of MIBG HMR to a medium-energy collimator condition [21] can be easily performed by using widely available computer software (smart MIBG software, Tokyo, Japan). When using standardized MIBG HMR data in the previous multicenter studies [18,19,20], this method identified identical prognostic values of HMR, contributing to a more extensive use of the MIBG index for the prediction of long-term outcomes. This study used a standardized HMR of MIBG activity. The standardization technique was established by the cross-calibration phantom experiments with various collimators available clinically in Japan and Europe [21,22]. The standardization can minimize differences in MIBG HMR originated in variations of low-energy, low-medium-energy, and medium-energy collimators. Furthermore, our recent study shows that this standardization method can be applied in cardiac event prediction models [23]. Prognostic values of cardiac autonomic innervation assessed by standardized HMR are also shown in HF patients with preserved or mid-range LVEF, as well as in HF patients with reduced LVEF [24]. Thus, the standardization technique in cardiac MIBG imaging can contribute to facilitating the wide-spread use of the neuroimaging in a clinical setting and in a multicenter study for precise identification and monitoring of life-threatening cardiac conditions.

### 4.3. Clinical Implications

The synergistic prognostic values of the neurohumoral, structural, and electromechanical remodeling’s demonstrated here are conducive to deepening understanding of HF pathophysiology and to the development of more effective management strategy of heart failure. Improvement in HF risk assessment by the presented method can greatly reduce under- and/or over-estimations of cardiac mortality risks, both of which might have hindered clinically appropriate and cost-effective selection of HF management [25]. This includes therapies using devices such as ICD, CRT, and CRTD and transcatheter mitral valve repair, which have been shown to incrementally improve prognosis beyond standard neurohormonal treatments in a certain subgroup of HF patients. The integration of LVMD and cardiac sympathetic function assessment into conventional prognostic parameters might be helpful for selecting HF patients at increased risk who can benefit more definitively from advanced device therapy. Recently, we developed a machine learning (ML)-based risk model using ^123^I-MIBG HMR to differentially predict cardiac death mode (pump failure death versus sudden/arrhythmic death) in HF patients [23]. The ML method can easily analyze various sorts of clinical, cardiac functional, and autonomic functional variables to identify the best model for risk-stratification and subsequent selection of the therapeutic strategy. Thus, a better HF risk-based approach has the potential for establishing a clinically reasonable health-care management strategy for treatment and prevention of unfavorable clinical outcomes, while corresponding appropriately to growing medical costs.

### 4.4. Study Limitations

Despite the entry and exclusion criteria in this study, the possibility of selection bias cannot be completely ruled out. This is because of the features such as an observational cohort study. Patients with systolic dysfunction who had undergone imaging studies for assessment of LVMI, LVMD, and cardiac sympathetic innervation were analyzed. It is necessary to establish better risk-based prophylactic and therapeutic strategies for patients who were identified to be at more increased risk for cardiac mortality by our method. It is also necessary to clarify effects of novel drugs recently available on LVH, LVMD, and cardiac sympathetic innervation in HF patients at greater risks. This study did not include any patients who had major valvular heat diseases responsible for advanced heart failure such as aortic stenosis (AS) or mitral regurgitation (MR), who may had been indicated for transcatheter aortic valve implantation (TAVI) or Mitra Clip^®^ treatment. Because of the possibilities of cardiac MIBG imaging for identifying the therapeutic effects of TAVI or Mitra Clip^®^ [26,27], a future study will clarify the therapeutic effects on cardiac sympathetic innervation and function in correlation with long-term clinical outcomes. Despite the possible prognostic values of right heart information in left-sided heart failure [28,29], there was no right-heart data available in all patients of this study. Besides technical difficulties in the assessment particularly in patients with advanced heart failure and left ventricular dilatation, the presented study aimed to correlate cardiac MIBG acidity quantified as standardized HMR with left ventricular functional parameters. Cardiac MIBG tomographic imaging enables identifying regionally denervated but viable myocardium. However, as patients with advance heart failure often have profound and global impairment of cardiac sympathetic innervation, it is actually impossible not only to visually identify cardiac MIBG activity but also to reconstruct cardiac MIBG activity using tomographic data. Therefore, MIBG tomographic data were not obtained in all patients enrolled for this study. Finally, cost-effective analysis is necessary when both myocardial gated perfusion SPECT and cardiac MIBG studies are performed in HF patients. In addition to the assessment of myocardial ischemia, viability, and left ventricular function, this study clearly showed that conventional myocardial perfusion gated SPECT study can be utilized to provide novel biomarkers that improve the risk-assessment of HF patients. Importantly, the quantitative information on LVH and LVMD presented here was obtained without any additional cost or time in a standard gated SPECT protocol. These findings indicate the possibilities that this method can not only improve the cost-effectiveness of a standard gated SPECT study per se, but also rationalize a total management cost in HF patients by reducing unnecessary or ineffective interventional approaches by more precise risk-stratification in HF patients. In this context, further study is needed to establish a more appropriate risk-based strategy using the imaging modalities from a prognostic and economic perspective.

## 5. Conclusions

Impairment of cardiac sympathetic innervation as well as three-dimensionally assessed left ventricular hypertrophy and mechanical dyssynchrony have critical prognostic values. The independent and synergistic interactions of these biomarkers on long-term outcomes indicate not only pathophysiological implications in the development and progression of heart failure, but also the clinical significance to help physicians more appropriately risk-stratify and select better management strategy in HF patients.

## Figures and Tables

**Figure 1 jcm-10-05047-f001:**
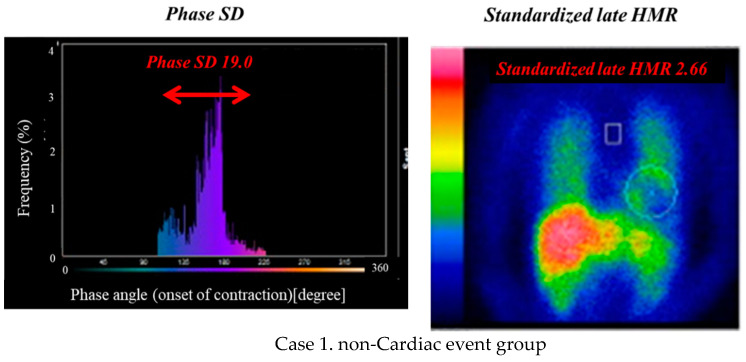
Phase histograms for measurements of phase SD (left panels) and heart/mediastinum ratio (HMR) of MIBG activity (right panels). Case 1: a 72-year-old female with ischemic cardiomyopathy who had reduced LVEF (32%), but had nearly normal values of LV mass index (97 g/m^2^), phase SD (19), and MIBG HMR (2.66). No cardiac event occurred during the follow-up period. Case 2: a 68-year-old male undergoing cardiac re-synchronization therapy who had non-ischemic cardiomyopathy with reduced LVEF (28%), increased LV mass index (178 g/m^2^), increased phase SD (40), and decreased MIBG HMR (1.48). He died of progressive pump failure during the follow-up period.

**Figure 2 jcm-10-05047-f002:**
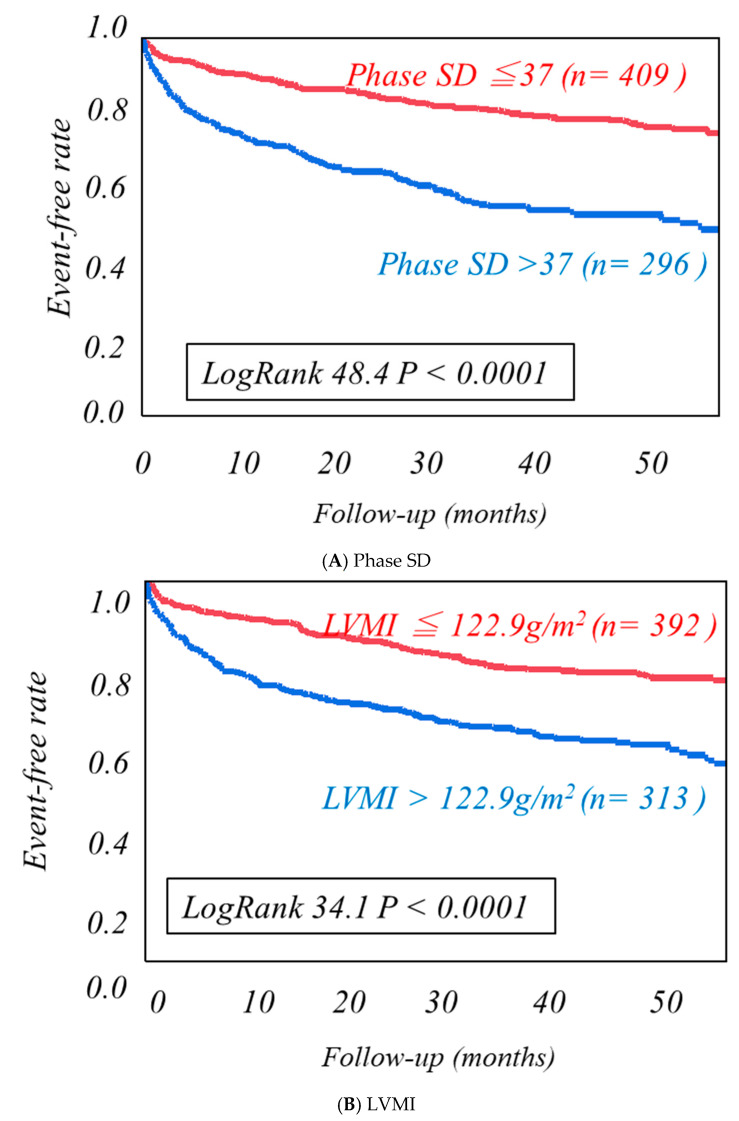
Kaplan−Meier event-free analysis of Phase SD and LVMI in three dimensions, standardized late HMR. Kaplan−Meier event-free curves clearly discriminate the high-risk patients from the low-risk patients using cutoff values determined by the ROC analysis, including phase SD of 36 degrees (**A**), left ventricular mass index (LVMI) of 122.9 g/m^2^ (**B**), and a MIBG heart/mediastinum ratio (HMR) of 1.89 (**C**).

**Figure 3 jcm-10-05047-f003:**
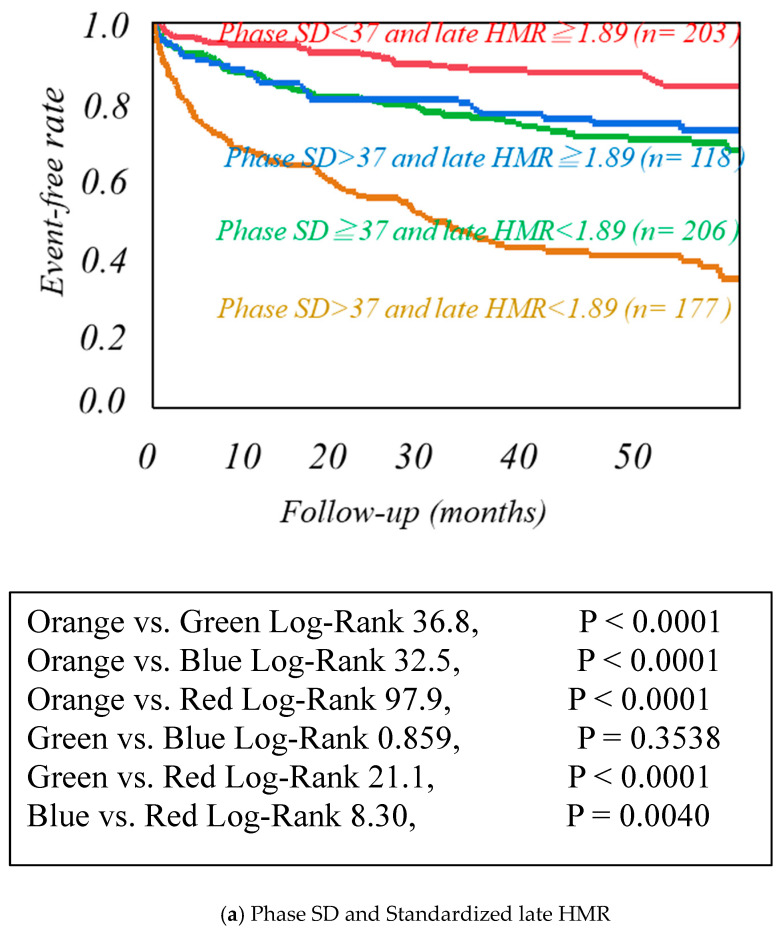
Kaplan−Meier event-free analysis of the combination of Phase SD and LVMI in three dimensions, standardized late HMR. Kaplan−Meier event-free curves created by the combinations of two of the three prognostic variables such as phase SD, left ventricular mass index (LVMI), and standardized MIBG heart/mediastinum ratio (HMR).

**Figure 4 jcm-10-05047-f004:**
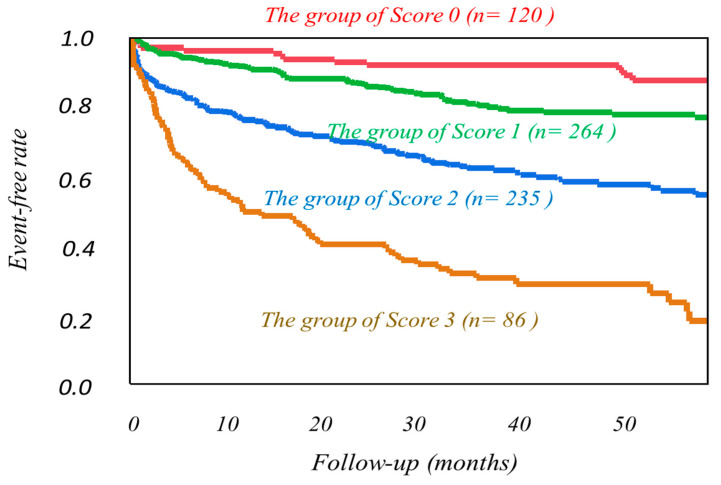
Kaplan−Meier event-free curves based on the accumulated number of the three prognostic variables such as phase SD, left ventricular mass index (LVMI), and standardized MIBG heart/mediastinum ratio (HMR) in overall patients. Phase SD and LVMI in three dimensions cut-off value determined by the ROC analysis for the prediction of cardiac events combination with three parameters.

**Figure 5 jcm-10-05047-f005:**
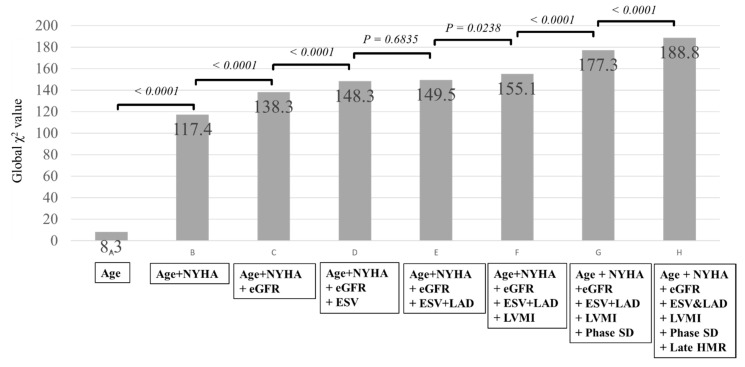
Additions of LVMI, phase SD, and MIBG HMR to the significant multivariate variables, including conventional clinical and cardiac functional biomarkers (Table 3), show significant increases in prognostic values, leading to a maximal Chi-square value of 188.8. Abbreviations are the same as those in Table 1 and Table 2.

**Table 1 jcm-10-05047-t001:** Comparison of clinical data between groups with and without cardiac events.

	Cardiac Events Group (*n* = 246)	Non-Cardiac Events Group (*n* = 459)	*p*-Value
Age (years)	70.8 ± 11.4	65.1 ± 12.2	*p* < 0.0001
Gender (male/female)	189/57	328/131	ns
NYHA (I/II/III/IV)	85/76/74/11	410/38/6/5	*p* < 0.0001
Past history			
Hypertension	127 (51.6%)	224 (48.8%)	ns
Diabetes mellitus	86 (34.9%)	167 (36.3%)	ns
Dyslipidemia	87 (35.3%)	191 (41.6%)	ns
Atrial fibrillation	88 (35.7%)	127 (27.6%)	ns
Ventriculartachycardia/ventricular fibrillation	59 (23.6%)	64 (13.9%)	*p* = 0.0406
Hemodialysis	56 (22.6%)	58 (12.6%)	*p* = 0.0259
Etiology			
Ischemic	125 (50.8%)	204 (44.4%)	ns
Previous MI	97 (39.4%)	157 (34.2%)	ns
Post PCI	98 (39.8%)	164 (35.7%)	ns
Post CABG	51 (20.7%)	54 (11.7%)	*p* = 0.0020
Device implantation			
ICD implantation	29 (11.7%)	48 (10.4%)	ns
CRT implantation	21 (8.5%)	37 (8.1%)	ns
Laboratory data			
Hemoglobin (g/dL)	11.6 ± 2.1	12.2 ± 2.2	*p* < 0.0001
eGFR (mL/min/1.73 m^2^)	35.6 ± 26.4	50.0 ± 27.9	*p* < 0.0001
Sodium (mmol/L)	139.0 ± 4.3	139.8 ± 3.6	*p* = 0.0139
BNP/NTproBNP staging (I/II/III/IV)	6/11/17/212	30/54/78/297	*p* < 0.0001
Medication			
ACE-I/ARB	138 (56.1%)	274 (59.6%)	ns
β-blockers	226 (91.8%)	424 (92.3%)	ns
Loop diuretics	185 (75.2%)	358 (77.9%)	ns
Mineralocorticoid receptor antagonist	71 (28.8%)	139 (30.2%)	ns
Anti-vasopresin agents	50 (20.3%)	53 (11.5%)	*p* = 0.0027
Calcium channel blockers	55 (22.3%)	133 (28.9%)	ns
Nitrates	38 (15.4%)	53 (11.5%)	ns
Amiodarone	97 (39.4%)	102 (22.2%)	*p* < 0.0001
Statins	72 (29.2%)	195 (42.4%)	*p* = 0.0005

Values are shown as means ± one standard deviation, MI—myocardial infarction; PCI—percutaneous coronary intervention; CABG—coronary artery bypass grafting ; ICD—implantable cardioverter-defibrillator ; CRT—cardiac resynchronization therapy ; eGFR—estimated glomerular filtration rate ; NYHA—New York Heart Association Classification; ACE-I—angiotensin-converting enzyme-inhibitors; ARB—angiotensin-receptor blockers; ns—no significance.

**Table 2 jcm-10-05047-t002:** Comparison of two-dimensional echocardiographic parameters and phase SD and LV weight and LVMI of rest ^99m^Tc, the finding of MIBG imaging between groups with and without cardiac events.

	Cardiac Events Group (*n* = 246)	Non-Cardiac Events Group (*n* = 459)	*p*-Value
M-mode			
LVDd (mm)	56.2 ± 11.4	55.1 ± 8.9	ns
LVDs (mm)	47.1 ± 12.4	44.6 ± 9.8	*p* = 0.0042
LAD (mm)	44.0 ± 8.0	40.8 ± 6.8	*p* < 0.0001
Modified Simpson method			
LVEF (%)	33.0 ± 11.8	36.7 ± 9.8	*p* = 0.0016
EDV (ml)	163.4 ± 74.5	152.0 ± 54.8	*p* = 0.0244
ESV (ml)	112.3 ± 67.3	96.4 ± 48.4	*p* = 0.0004
Doppler method			
E wave velocity (m/sec)	0.87 ± 0.28	0.81 ± 0.28	*p* = 0.0108
Dct (msec)	188.6 ± 96.8	194.8 ± 76.4	ns
Tissue Doppler method			
Septal E/e′	20.2 ± 8.0	17.2 ± 6.7	*p* < 0.0001
The findings of 12-lead electrocardiogram			
QRS complex duration (msec)	126 ± 30	120 ± 29	*p* = 0.0033
LBBB (%)	15 (6.1%)	20 (4.3%)	ns
RBBB (%)	28 (11.3%)	30 (6.5%)	ns
Biventricular pacing (%)	21 (8.5%)	37 (8.1%)	ns
The findings of Rest ^99m^Tc			
Phase SD	37.4 ± 10.8	33.1 ± 9.9	*p* < 0.0001
Left ventricular weight(g)	204.6 ± 51.4	189.9 ± 42.2	*p* < 0.0001
Left ventricular mass index (LVMI)(g/m^2^)	129.3 ± 35.7	117.9 ± 28.7	*p* < 0.0001
The findings of MIBG imaging			
Washout Ratio	29.3 ± 9.9	27.5 ± 9.8	*p* = 0.0294
Standardized early HMR	2.03 ± 0.45	2.19 ± 0.44	*p* < 0.0001
Standardized late HMR	1.71 ± 0.39	1.99 ± 0.48	*p* < 0.0001

Values are shown as means ± one standard deviation. LAD—left atrial diameter; LV—left ventricular; LVEF—left ventricular ejection fraction; LVDd—end-systolic left ventricular diameter; EDV—left ventricular end-diastolic volume; ESV—left ventricular end-systolic volume ; Dct—left ventricular deceleration time; Left ventricular mass index = Left ventricular weight/BSA (m^2^); BSA (Du Bois formula) = Body Height^0.725^(m) × Body Weight^0.425^(kg) × 0.007184; RBBB—right bundle branch block; LBBB—left bundle branch block; ns—no significance.

**Table 3 jcm-10-05047-t003:** Results of the univariate and multivariate analyses.

	**Univariate Analysis**				
			**95% CI**		
	χ^2^	Hazard ratio	Lower	Upper	*p*-value
Age	35.1	1.033	1.012	1.046	<0.0001
NYHA functional class	216	2.715	2.412	3.049	<0.0001
Hemoglobin	16.7	0.888	0.840	0.940	<0.0001
Estimated GFR	46.2	0.983	0.979	0.988	0.0001
BNP/NTproBNP staging	35.6	1.683	1.5555	2.559	<0.0001
Amiodarone	30.8	2.130	1.648	2.751	<0.0001
Statins	10.1	0.647	0.489	0.848	0.0015
LAD	33.6	1.054	1.035	1.073	<0.0001
ESV	16.9	1.004	1.002	1.006	<0.0001
Septal E/e’	23.4	1.041	1.025	1.056	<0.0001
Phase SD	34.6	1.036	1.024	1.048	<0.0001
LVMI	25.6	1.010	1.006	1.014	<0.0001
Standardized late HMR	58.1	0.324	1.023	1.048	<0.0001
	**Multivariate Cox Proportional-Hazards Model Analysis**				
			**95% CI**		
	χ^2^	Hazard ratio	Lower	Upper	*p*-value
Age	4.83	1.013	1.001	1.026	0.0280
NYHA functional class	89.1	2.104	1.817	2.428	<0.0001
Hemoglobin	1.88	0.954	0.893	1.019	0.1694
Estimated GFR	33.5	0.984	0.978	0.989	<0.0001
BNP/NTproBNP staging	14.5	1.334	1.110	1.824	<0.0001
Amiodarone	13.2	1.587	1.278	2.218	0.0002
Statins					
LAD	4.59	1.020	1.001	1.040	0.0320
ESV	7.72	1.003	1.001	1.006	0.0054
Septal E/e’	2.29	1.001	0.995	1.030	0.1354
Phase SD	42.5	1.039	1.003	1.054	<0.0001
LVMI	9.37	1.006	1.002	1.010	0.0018
Standardized late HMR	36.9	0.397	0.291	0.539	<0.0001

## Data Availability

Data are available on reasonable request to the corresponding author.

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
