# Peer review of "Impaired Cardiac Sympathetic Innervation Increases the Risk of Cardiac Events in Heart Failure Patients with Left Ventricular Hypertrophy and Mechanical Dyssynchrony"

_jcm, 2021, doi:10.3390/jcm10215047_

Round 1

Reviewer 1 Report

This is a well written and well-presented article prospectively exploring the prognostic value of mechanical dyssynchrony  left ventricular hypertrophy and impaired cardiac sympathetic innervation as measured during SPECT imaging using novel analytic tools.

This paper is an extension of the authors' study:

Doi T, Nakata T, Yuda S, Hashimoto A. Synergistic prognostication of left ventricular hypertrophy and three-dimensional mechanical dyssynchrony in heart failure. ESC Heart Fail. 2020 Feb;7(1):361-370. doi: 10.1002/ehf2.12578. Epub 2020 Jan 21. PMID: 31965750; PMCID: PMC7083410.

 and the study

Doi T, Nakata T, Yuda S, Hashimoto A. Synergistic prognostic implications of left ventricular mechanical dyssynchrony and impaired cardiac sympathetic nerve activity in heart failure patients with reduced left ventricular ejection fraction. Eur Heart J Cardiovasc Imaging. 2018 Jan 1;19(1):74-83. doi: 10.1093/ehjci/jew334. PMID: 28158459.

This study adds the aspect of the assessment of sympathetic innervation, to the assessment, suggesting that the combination of abnormal values of all three factors is associated with poor prognosis.

As the sympathetic innervation assessment is the major contribution of this manuscript, I would suggest that the introduction be significantly shortened, with the focus on this aspect and this addition, rather than diverging into CRT and heart failure in general. A more thorough description of the basis for the technique would be beneficial in the intro.

In the methods section, for each of the modalities, there should be some reference to how the measure was validated.

it is important to clarify whether there was any overlap in patients compared to the study

In the results section, . The figures comparing <65 and >65 should be placed next to each other to enable comparison and similarly the gender comparison. The issue being compared should appear in large letters rather than somewhat removed in the fine print, to enable the reader to rapidly understand the intent of the graph rather than searching for it.  Similarly, graph legends should refer to the group and not to the color, and be situated in a complementary manner.  The scoring should be better explained in the text, as readers are left somewhat at a loss. The ROC analysis should either be referenced to the ESC HF paper or presented as a supplement. 

The discussion should focus on the sympathetic innervation technique,and explore the literature in that directiion.

Reviewer 2 Report

Thank you for the opportunity to review this paper.

It is a well constructed manuscript with clinical relevance. The style is easy to follow and attempts to identify plausible biologic markers of poor outcomes in heart failure. I have a few questions regarding aspects of the paper.

  1. Myocardial perfusion SPECT is expensive, involves radiotion exposure and is not readily available in many centres globally. While the results of this paper may provide a hypothesis for the pathophysiology of heart failure, the tone of the paper is of clinical benefit in risk stratification.  Based on figure 5, Spect derived LVMI a, Phase SD and late HMR  provide modest incremental benefit in risk stratification. To prove clinical utility, could the authors comment on whether this increase is significantly greater than what could be provided with ECG and echocardiography - ie traditional markers of LV mass (echo derived) and dyssynchrony (ECG derived or echo derived). At mininum LV mass on echo and LBBB on ECG could be compared.

2. A fair limitation recognised by the paper is selection bias. The results presented hint that selection bias may be a more significant issue than recognised, with a higher proportion of patients in the event group receiving amiodarone. This may reflect local practice. What was the indication for amiodarone use? Was it ventricular arrythmias (which may explain the worse outcomes independently) or atrial arrhythmias, and was there a difference in indication between groups? Could amiodarone be an explanation for worse outcomes?

3. There are too many figures. Repeating the curves for different sub-groups does not add to the paper, and could be in an appendix or supplementary file.

Overall, this is an interesting paper but leaves much unexplored. To be able to draw clinical conclusions I recommend the authors attempt to demonstrate this technique is superior to standard of care, otherwise the focus should be on the pathophysiology and potential to guide therapies - hypothesis generation.
